# SPLICING UP YOUR PREDICTIONS WITH RNA CONTRASTIVE LEARNING

## ABSTRACT

In the face of rapidly accumulating genomic data, our understanding of the RNA regulatory code remains incomplete. Recent self-supervised methods in other domains have demonstrated the ability to learn rules underlying the data-generating process such as sentence structure in language. Inspired by this, we extend contrastive learning techniques to genomic data by utilizing functional similarities between sequences generated through alternative splicing and gene duplication. Our novel dataset and contrastive objective enable the learning of generalized RNA isoform representations. We validate their utility on downstream tasks such as RNA half-life and mean ribosome load prediction. Our pre-training strategy yields competitive results using linear probing on both tasks, along with up to a two-fold increase in Pearson correlation in low-data conditions. Importantly, our exploration of the learned latent space reveals that our contrastive objective yields semantically meaningful representations, underscoring its potential as a valuable initialization technique for RNA property prediction.

## 1 INTRODUCTION

Mature RNAs are molecules that encode genetic information and are thoroughly regulated by the cell to control protein expression and other functions. Many aspects of this regulation are determined by the RNA sequence. Experimental procedures measuring these properties have been instrumental in understanding cellular function and disease impact. However, experiments are often high-cost and time-consuming. Supervised learning models trained on genetic sequences to predict cellular function provide effective, low-cost tools. Computational methods have been applied to model cellular processes such as splicing, the process of RNA assembly, and RNA half-life prediction (Jaganathan et al., 2019; Linder et al., 2022; Agarwal & Kelley, 2022). These models can be used to identify disease mechanisms (Merico et al., 2020; Richards et al., 2015), improve therapeutics such as messenger RNA (mRNA) vaccines (Celaj et al., 2023), and serve as low-cost tools for drug design (Wainberg et al., 2018). Despite the importance of these applications, the difficulty associated with experimental data acquisition restricts training supervised methods for a wider range of tasks.

In recent years, techniques from self-supervised learning (SSL) have enabled generation of effective representations that can be fine-tuned on related downstream tasks. This has reduced reliance on labeled data and demonstrated impressive generalization capabilities to a diversity of tasks (Tomasev et al., 2022; Radford et al., 2021). SSL can be formulated through a data reconstruction objective, where a model is required to reconstruct a portion of the input data. Typical formulations have included next token prediction (NTP) and masked language modeling (MLM) (Devlin et al., 2018; Radford & Narasimhan, 2018; Vaswani et al., 2017). In genomics, many recent self-supervised methods such as Ji et al. (2021); Chen et al. (2022); Nguyen et al. (2023) use this SSL paradigm. However, there are unique properties associated with genomic data that introduce challenges to applying a reconstruction-based SSL objective or supervised learning.

Genomic sequences in the natural world are constrained by evolutionary viability, resulting in low natural diversity[1] and high mutual information between samples. Only two percent of the human genetic code is translated into protein and can be considered high information content. The remaining 98% of the genetic sequence is under no or very little negative selection, meaning mutations have a small impact on organism fitness. Without a strong biological inductive bias, existing

---

[1]In the coding region, an average individual carries $27 \pm 13$ mutations(Lek et al., 2016; Taliun et al., 2021).

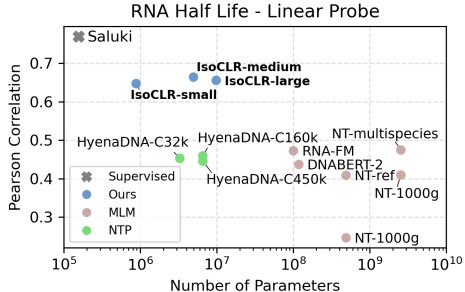 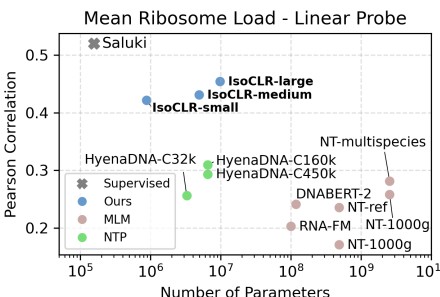

Figure 1: Pearson correlation of linear regressions trained on representations by different self-supervised methods. RNA half-life and mean ribosome load are important cellular properties for regulating protein abundance. IsoCLR's contrastive objective outperforms existing SSL approaches.

reconstruction-based SSL models often reconstruct non-informative tokens, which can result in suboptimal representations. Due to the high-mutual information between samples, it is also difficult to scale the effective size of the training dataset to circumvent this issue. We find that recent applications of SSL methods to genomics (Dalla-Torre et al., 2023; Ji et al., 2021; Nguyen et al., 2023) learn latent representations that are not well linearly separated, as visualized in Figure 1. The gap between baseline SSL methods and supervised approaches remains large, while no clear trend exists between model size and performance.

Genomics data is also susceptible to distribution shifts that arise from differences in experimental protocols. Measurements for the same cellular process can have low correlations (Agarwal & Kelley, 2022). Supervised approaches can overfit experimental-specific signals, particularly in low-data situations. However, effective SSL models that capture the underlying biology can enable few-shot learning by fine-tuning with just a few samples from a new experimental procedure.

In this work, we develop IsoCLR, a contrastive technique applied to genomic data with the purpose of learning effective RNA representations. Contrastive learning, a type of self-supervised learning, utilizes data augmentations to alter samples in a semantically insignificant way to learn effective representations (Koch, 2015; Chen et al., 2020). IsoCLR utilizes stronger biologically motivated inductive biases, making it less reliant on limited sequence diversity and more capable of learning representations without training on experimental data (Figure 2). To generate RNA augmentations, we rely on naturally occurring cellular and evolutionary processes: alternative splicing, and gene homology. Byproducts of these processes generate RNAs with different sequences and similar functions (Pertea et al., 2018). We identify paired RNA sequences generated by these processes and use them as augmentations for learning RNA embeddings. By minimizing the distance between functionally similar sequences, the model can learn regulatory regions critical for RNA property and function prediction. Please refer to A.1 section for a description of these processes.

We pre-train a dilated convolutional residual model which has been demonstrated to be successful in applications for cellular property prediction by being able to generalize to long variable length sequences (Kelley et al., 2018; Linder et al., 2022; Chen et al., 2016; He et al., 2015). Utilizing biologically inspired RNA augmentations allows us to generate robust multi-purpose RNA representations. We investigate the effectiveness of these representations by evaluating the models on RNA half-life and mean ribosome load tasks (Agarwal & Kelley, 2022; Sugimoto & Ratcliffe, 2022). We find that IsoCLR outperforms all the other self-supervised methods and matches supervised performance when fine-tuning. Our main contributions are:

- We create a novel pre-training dataset by proposing augmentations for genomic sequences produced through homology, and alternative splicing processes.

- We propose IsoCLR, a novel method that employs a contrastive learning objective to learn robust RNA isoform representations across species.

- We conduct extensive evaluations of IsoCLR on tasks such as RNA half-life and mean ribosome load prediction. Our results demonstrate improvements, particularly in the low data regime.

Figure 2: Description of the data generation and training processes for IsoCLR. The **upper** half of the figure demonstrates hypothetical examples for creating mature RNA sets from which positive data pairs are sampled. The first example demonstrates that a positive mature RNA set can constructed from splicing. The second example demonstrates RNA set construction from gene homology. The **lower** half of the figure demonstrates the training process utilizing the generated mature RNA sets. First RNAs are sampled with replacement from the sets and an RNA embedding is generated using a dilated convolutional residual encoder $f$. Then the representations are passed through a projector $g$, the normalized output of which is used to compute the decoupled contrastive loss.

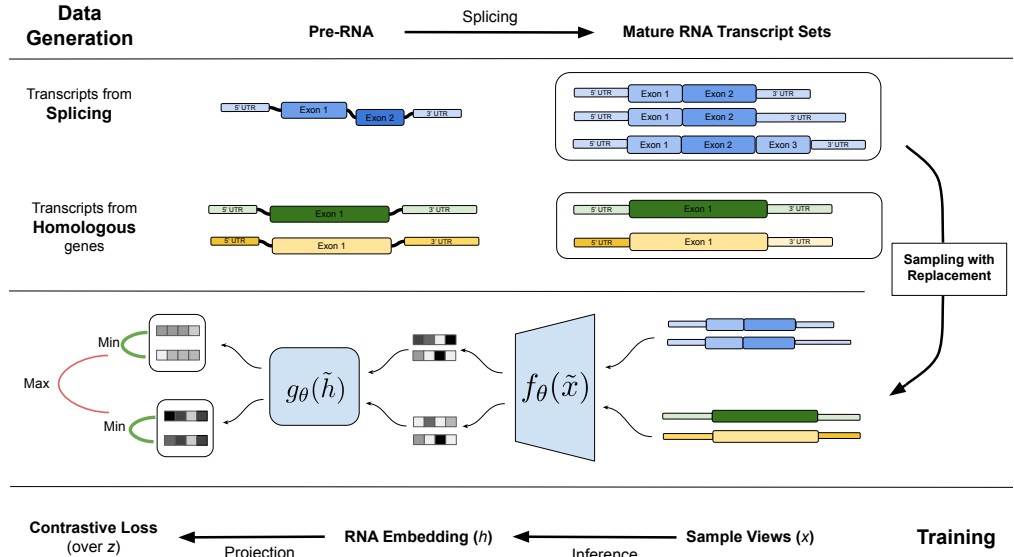

## 2 RELATED WORKS

This work builds on top of foundational efforts spread across three main areas: contrastive representation learning, self-supervised applications in cellular property prediction, and methods for enriching genetic sequence input beyond one hot encoded representation.

**Contrastive methods.** We build the IsoCLR approach for RNA sequences utilizing a rich body of work exploring contrastive learning for computer vision (Balestriero et al., 2023). A fundamental deep metric learning approach is SimCLR in which the authors propose minimizing the representation distance between two views from the same sample while maximizing the distance between views from different samples (Chen et al., 2020). This approach does not require labeled data and is based on the availability of domain-specific augmentations. Methods like BYOL and VicReg followed and were able to reformulate the contrastive approach by removing the need for in-batch negative samples (Grill et al., 2020; Bardes et al., 2021). They propose solutions to the trivial solution collapse problem through a variance regularization loss term and architectural design choices. Recent work aims to unify these methods under the contrastive formulation by making a distinction between *sample* and *dimension* contrastive methods (Garrido et al., 2022).

**Self-supervised learning for cellular properties.** Due to the common sequence-based representation between genomics and language, self-supervised learning techniques have long been explored in genomic sequence property predictions. DNABert utilized the BERT problem formulation to learn an encoding for 500 nucleotide long sequences and demonstrated the value for splice site predictions and other tasks (Ji et al., 2021; Devlin et al., 2018; Zhou et al., 2023). Nucleotide Transformer (NT), another masked language modeling method, demonstrated the utility of doing data collection from multiple species (Dalla-Torre et al., 2023). RNA-FM was trained to predict non-coding RNA properties with masked language modeling using 23 million non-coding sequences (Chen et al., 2022). Recently, HyenaDNA has demonstrated that applying long convolutions replacing the attention operation, can lead to effective DNA property prediction while scaling the input sequence length to a million tokens (Nguyen et al., 2023). In the distinct protein representation learning space, there is a

variety of protein language models utilizing auto-regressive and masked language modeling losses to predict protein properties like structure, variant effects, and functional properties (Meier et al., 2021; Lin et al., 2023). Contrastive learning has also been used in more specialized domains such as enzyme property prediction while utilizing known shared enzyme properties as views of similar sequences (Yu et al., 2023). Contrastive methods have also been used to learn a more general representation of protein function by maximizing the mutual information between global and local sequence representations (Lu et al., 2020). We build on these works by exploiting domain-specific RNA augmentation to build general representations that are architecture-agnostic.

**Beyond one hot encoded genomes.** Another important area for advancing cellular property prediction is iterating beyond the reference genome for representing genomic sequences. One such strategy is to integrate random biologically plausible augmentations during training (Lee et al., 2023). By using domain-specific knowledge of the types of augmentations introduced during evolutionary processes, the authors demonstrate they can improve the performance of supervised models for predicting DNA properties. Using multiple sequence alignments is another way to use homology information, common in the protein modeling space (Do et al., 2005; Frazer et al., 2021; Jumper et al., 2021). In another perspective, authors have argued that evolutionary homologs are a viable path for generating augmentations (Lu et al., 2020). In the RNA space, the authors of Saluki demonstrated that additional information indicating the locations of splice sites and coding sequences can help learn RNA properties (Agarwal & Kelley, 2022). In this work, we integrate augmentation, homology, and additional input concepts to produce general RNA representations.

## 3 METHODS

### 3.1 CONTRASTIVE LEARNING DATASET

Our proposed dataset used for the contrastive learning objective is composed of annotated mRNA transcriptomes. Gencode and Refseq databases compile mRNA isoforms for different species, indicating relative positions of exonic coordinates and other important genomic features such as 5'UTR, CDS, and 3'UTR regions (Frankish et al., 2021; O'Leary et al., 2016). Using this information, we generated a six-track mature RNA representation, consisting of four one-hot encoded tracks representing genomic sequence, a track indicating the 5' location of splice sites, and a track indicating the first nucleotide of every codon. The addition of extra tracks indicating splice site and coding sequence locations has been shown to be beneficial for downstream genomic tasks (Agarwal & Kelley, 2022). Depending on the species analyzed and the transcriptome annotation resource used, between 25% and 50% of genes contain multiple isoforms which we then sample to use as augmentations. Additionally, we used homology as a source of RNA isoform invariances. Homologous genes, which share structural similarities and encode similar functions, can be divided into paralogous and orthologous relationships. Paralogs result from gene duplication events, while orthologs result from evolutionary speciation events where species have similar genes due to sharing a common evolutionary ancestor (Lesk, 2020). To annotate these relationships, we used the Homologene database (Sayers et al., 2023).

### 3.2 CONTRASTIVE LEARNING OBJECTIVE

In the vision domain, contrastive learning strategies have had significant success by identifying augmentations that do not have a strong semantic effect, such as cropping, rotation, or Gaussian blur (Yun et al., 2019; Zhang et al., 2017; Chen et al., 2020). In this work, we use RNA splicing isoforms and homologous genes as sources for sequence functional invariance. By sampling RNA isoform

| # Species | # Genes | # Transcripts | Mean # Transcripts | % Genes with $> 2$ Transcripts |
|---|---|---|---|---|
| 10 | 228,800 | 926,628 | 4.0 | 29% |
| 2 | 65,600 | 286,390 | 4.36 | 41% |
| 1 | 42,800 | 222,492 | 5.19 | 51% |

Table 1: Descriptive statistics for the contrastive learning dataset. As we utilize more species for dataset construction the number of sequences grows.

sequences produced from the same gene, we are able to generate sequence variation without significantly changing the functional properties of the molecule. Similarly, with homologous genes, we are able to pool RNA transcripts from evolutionarily related genes and generate sequence diversity without altering transcript function (Pertea et al., 2018).

During our contrasting training phase, we pool together sequences of splicing isoforms from homologous genes and treat them as views of the same object. Given a batch of $N$ sequences (e.g. RNA isoforms) $x_1, ...x_N$ let $x_i^1$, $x_i^2$ be two splicing isoforms coming from a set of homologous genes. We pass these augmented views through a dilated convolutional encoder $f$ resulting in the outputs $h_i^1$ and $h_i^2$. These representations are then fed into a multi-layer perceptron projection head, $g$ the output of which is used to calculate normalized projections $z_i$ as shown in figure 2:

$$z_i^1 = \frac{g(h_i^1)}{\|g(h_i^1)\|} \text{ and } z_i^2 = \frac{g(h_i^2)}{\|g(h_i^2)\|}. \tag{1}$$

Normalized projections $z_i$ are used to compute the decoupled contrastive loss, utilizing samples from the rest of the batch as negatives (Yeh et al., 2021).

We utilize the decoupled contrastive loss (DCL) for our contrastive objective as it has been shown to require smaller batch sizes, is less sensitive to other hyperparameters such as learning rate, and the positive loss term can be weighted by difficulty (Yeh et al., 2021). DCL iterates on the normalized temperature-scaled cross-entropy loss by splitting the contrastive objective into two terms: a similarity loss and a dissimilarity loss (Sohn, 2016). The cost function can be formulated as the sum of two terms where $z_i^1$ and $z_i^2$ are two views generated from the set of homologous genes while $z_j$ are sampled from different sets of homologous genes. More formally, the positive and negative losses are calculated in the following way:

$$\mathcal{L}_{DCL,i}(\theta) = \log \sum_{z_k \in \mathcal{Z}, l \in 1,2}^{N} \mathbb{1}_{k \neq i} \exp(\langle z_i^1 \cdot z_k^l \rangle / \tau) - w_i \langle z_i^1, z_i^2 \rangle / \tau. \tag{2}$$

In the above $\tau$ is the temperature parameter set to $0.1$, and $\mathbb{1}_{k \neq i}$ is an indicator function that evaluates to 1 when $k \neq i$. Due to the non-uniform number of views per set of homologous genes, we use the term $w_i$ to up-weight the importance of difficult examples containing more transcripts and down-weight the importance of the positive loss when a gene has only a single transcript. The above loss is computed for all the samples in the batch for both the sampled views $l \in 1, 2$.

Given that the views in our objective are naturally occurring transcripts, sets of homologous genes will have a different number of splicing isoforms to sample from for the contrastive objective. Many non-protein coding genes will have only a single splicing isoform, resulting in the two views being identical. To make the positive objective of identifying the augmented isoform non-trivial, we use dropout in our model and, randomly mask 15% of the transcript sequence (Ji et al., 2021). This enforces the positive loss term for samples with a single RNA sequence to be non-zero. However, samples with multiple sequences generated through splicing and homology processes are more informative to the model. To reflect this unequal difficulty between samples in our positive loss term, we introduce a weighting term $w_i$ to increase the importance of samples with a higher number of splicing isoforms:

$$w_i = log(t_i + c)\frac{T}{\Sigma_{k=1}^{N} log(t_k + c)}, \tag{3}$$

Where $t$ is the number of transcripts per gene set, $T$ corresponds to the total count of transcripts in the dataset, and $c$ is a constant. The above objective increases the importance of samples with multiple RNA views while maintaining the overall norm of the total loss at the start of training. The weighting is applied only to the positive loss since the negative loss responsible for maximizing the distance between different samples is not affected by the number of transcripts per sample.

### 3.3 DOWNSTREAM TASKS

**RNA half-life** (RNA HL) is an important cellular property to measure due to its implications for human disease. A genetic mutation in the non-coding region of an RNA can result in increased

RNA turnover, in turn decreasing protein abundance. Recently, it has been shown that the choice of method for measuring RNA half-life can have an outsize impact with no clear ground truth (Agarwal & Kelley, 2022). To address this problem, Agarwal and Kelley (2022) utilized the first principal component of over 40 different RNA half-life experiments. The dataset consists of 10,432 human and 11,008 mouse RNA sequences with corresponding measurements. The low data availability and high inter-experiment variation underscore the importance of data efficiency, and generalizability in computational models to be developed for this task. In addition, the authors find that augmenting the input tracks of the model beyond one hot encoded sequence to include a coding sequence, and a splice site track significantly increased performance. We utilize the constructed RNA half-life dataset for evaluating the effectiveness of IsoCLR representation.

**Mean ribosome load** (MRL) is a measure of the translational efficiency of a given mRNA molecule. It measures the number of ribosomes translating a single mRNA molecule at a point in time. Accurate MRL measurement is crucial as it offers insights into the efficiency of protein translation, a key process in cellular function. The dataset in question, derived from the HP5 workflow, captures this metric across 12,459 mRNA isoforms from 7,815 genes (Sugimoto & Ratcliffe, 2022). This dataset was derived from a single experiment so we can expect a higher amount of noise associated than the RNA half-life dataset.

**Gene ontology** (GO) terms are a hierarchical classification system used for assigning function to genes and their products (Consortium et al., 2023; Ashburner et al., 2000; Zhou et al., 2019). In this work, we utilize GO classes to visualize model latent embeddings. GO terms are generated through biological experiments, dedicated to measuring specific molecular properties such as GO:0005102, a term corresponding to signaling receptor binding. The hierarchical systems allow for fine-grained annotation of function, with broader terms at the top of the hierarchy and increased specificity closer to the bottom. To annotate genes with gene ontology terms, we subset GO classes three levels from the root labeling all available genes.

### 3.4 IMPLEMENTATION DETAILS

We use a dilated convolutional residual network for our encoder $f$ with exponentially increasing dilation (Chen et al., 2016). For most experiments, we use a model with 8 residual blocks, each followed by a max pooling layer. For the projection head used only during contrastive pre-training, we use a three-layer MLP with a hidden dimension of 2048 (Garrido et al., 2022). During contrastive training, we use weight decay of 1e-6. For both contrastive training and fine-tuning, we use the AdamW optimizer with 10 warm-up steps and cosine decay with a maximum learning rate of 0.01 (Kingma & Ba, 2014; Loshchilov & Hutter, 2016). For normalization during training, we use batch normalization and dropout (Ioffe & Szegedy, 2015; Srivastava et al., 2014).

## 4 EXPERIMENTAL RESULTS

We demonstrate that contrastive pre-training across homologous genes and splicing isoforms improves downstream prediction for both mRNA half-life evaluation and mean ribosome load estimation. We evaluate the effectiveness of the learned representation with three strategies: linear probing, full model fine-tuning, and latent space visualization. In addition, we highlight the effectiveness of pre-trained representations in low-data settings. For latent space visualization, we generate the results from the output of the encoder $f$ and visualize RNA representations utilizing the corresponding gene ontology categories. We demonstrate that our learned embedding has information on biological processes, cellular components, and molecular function without ever explicitly learning those categories.

### 4.1 ISOCLR LEARNS A GOOD LATENT SPACE REPRESENTATION

To evaluate the effectiveness of our pre-trained representation, we followed the conventional evaluation strategy of linear probing. If the learned latent representation is effective, then there $\exists\, \mathbf{w}$ s.t. $\mathbf{w}^T \mathbf{X} + b = \hat{y}$, where $\mathbf{X}$ is a matrix of embeddings and $\hat{y}$ approximates $y$. To evaluate the above, we freeze the weights of the dilated convolutional encoder $f$ and train a linear layer to predict RNA half-life and mean ribosome load tasks. Further experimental details are described in

Table 2: Linear probing results for self-supervised methods. The embeddings were computed for each method and then linear regression was computed analytically using the corresponding labels for each task. Bolded numbers indicate best performing model.

| Model Name | RNA HL Human MSE | RNA HL Human R | RNA HL Mouse MSE | RNA HL Mouse R | MRL MSE | MRL R |
|---|---|---|---|---|---|---|
| IsoCLR-S | 0.5860 | 0.6476 | 0.5843 | 0.6565 | 0.7594 | 0.4214 |
| IsoCLR-M | **0.5623** | **0.6647** | **0.5750** | **0.6633** | 0.7516 | 0.4312 |
| IsoCLR-L | 0.5736 | 0.6564 | 0.5845 | 0.6568 | **0.7329** | **0.4542** |
| DNA-BERT2 | 0.8417 | 0.4375 | 0.8304 | 0.3781 | 0.8442 | 0.2413 |
| NT-500m-1000g | 1.0280 | 0.2463 | 0.9826 | 0.2953 | 0.9718 | 0.1711 |
| NT-500m-human-ref | 0.8954 | 0.4093 | 0.8489 | 0.4006 | 0.9085 | 0.2356 |
| NT-2.5b-1000g | 0.8914 | 0.4099 | 0.8549 | 0.3745 | 0.9213 | 0.2583 |
| NT-2.5b-multi-species | 0.8862 | 0.4752 | 0.8807 | 0.4426 | 1.0015 | 0.2813 |
| Hyena-32K-seqlen | 0.8273 | 0.4532 | 0.7728 | 0.4449 | 0.8405 | 0.2566 |
| Hyena-160K-seqlen | 0.8053 | 0.4595 | 0.7876 | 0.4589 | 0.8061 | 0.2934 |
| Hyena-450K-seqlen | 0.7996 | 0.4462 | 0.7989 | 0.4602 | 0.8204 | 0.2934 |
| RNA-FM | 0.7849 | 0.4729 | 0.8305 | 0.4384 | 0.8854 | 0.2030 |

Appendix A.2. We demonstrate that IsoCLR outperforms other evaluated self-supervised methods on both tasks by a substantial margin in Figure 1 and Table 2.

We observe mixed results with regard to scaling the number of model parameters in terms of linear probing results. We see a clear improvement trend in MRL prediction, but we do not observe the same trend in terms of RNA half-life evaluation. Similarly, we observe that for other self-supervised models, the number of parameters does not improve performance. The clearest improvement trend we observe is in the Nucleotide Transformer work, where increasing the diversity of the training set by scaling the number of species improves performance. Similarly in our work, we aggregate highly informative sequences across 10 species. This demonstrates a path to further improve model effectiveness in genomic property prediction.

Finally, we evaluate whether IsoCLR's pre-trained representations capture fundamental biological functions. We examined high-level biological labels associated with individual genes. More specifically, we examined how well IsoCLR captures differences between gene-ontology terms associated with cellular components, biological processes, and molecular function (Figure 3a). We generate the representation with encoder $f$ and reduce the dimensionality of the embedding with t-sne (van der Maaten & Hinton, 2008). To quantitatively verify the latent structure, we perform linear probing over 10 GO classes and find they are linearly separable in the IsoCLR latent space. (Appendix A.4)

## 4.2 FINE TUNING AND DATA EFFICIENCY

To assess whether the IsoCLR pre-training objective provides utility beyond an effective representation, we evaluate its performance by fully fine-tuning it and comparing it to a supervised model

| Model Name | RNA HL Human MSE | RNA HL Human R | RNA HL Mouse MSE | RNA HL Mouse R | MRL MSE | MRL R |
|---|---|---|---|---|---|---|
| IsoCLR-S | **0.44 ± 1e-2** | **0.76 ± 8e-3** | **0.53 ± 3e-2** | **0.70 ± 1e-2** | **0.70 ± 3e-2** | 0.50 ± 2e-2 |
| IsoCLR-M | 0.48 ± 1e-2 | 0.74 ± 7e-3 | 0.56 ± 2e-2 | 0.69 ± 2e-2 | 0.76 ± 3e-2 | 0.49 ± 3e-2 |
| HyenaDNA-Tiny | 0.79 ± 2e-2 | 0.47 ± 9e-3 | 0.79 ± 6e-2 | 0.48 ± 2e-2 | 0.91 ± 2e-2 | 0.05 ± 2e-2 |
| HyenaDNA-Small | 0.78 ± 1e-2 | 0.46 ± 5e-3 | 0.78 ± 3e-2 | 0.48 ± 8e-3 | 0.91 ± 2e-2 | 0.04 ± 4e-2 |
| Supervised-S | 0.50 ± 1e-2 | 0.71 ± 8e-3 | 0.59 ± 5e-2 | 0.66 ± 3e-3 | 0.69 ± 6e-2 | 0.50 ± 5e-2 |
| Supervised-M | 0.53 ± 3e-2 | 0.63 ± 3e-2 | 0.64 ± 8e-2 | 0.69 ± 2e-2 | 0.82 ± 6e-2 | 0.43 ± 6e-2 |
| Saluki | **0.44 ± 1e-2** | **0.76 ± 1e-2** | 0.55 ± 5-e2 | **0.70 ± 3e-2** | 0.67 ± 4e-2 | **0.52 ± 2e-2** |

Table 3: Mean squared error and Pearson correlations (R) of full model fine-tuning on RNA half-life (for human and mouse datasets) and mean ribosome load tasks. Best models shown in bold. Confidence intervals were computed using standard deviation over three random seeds.

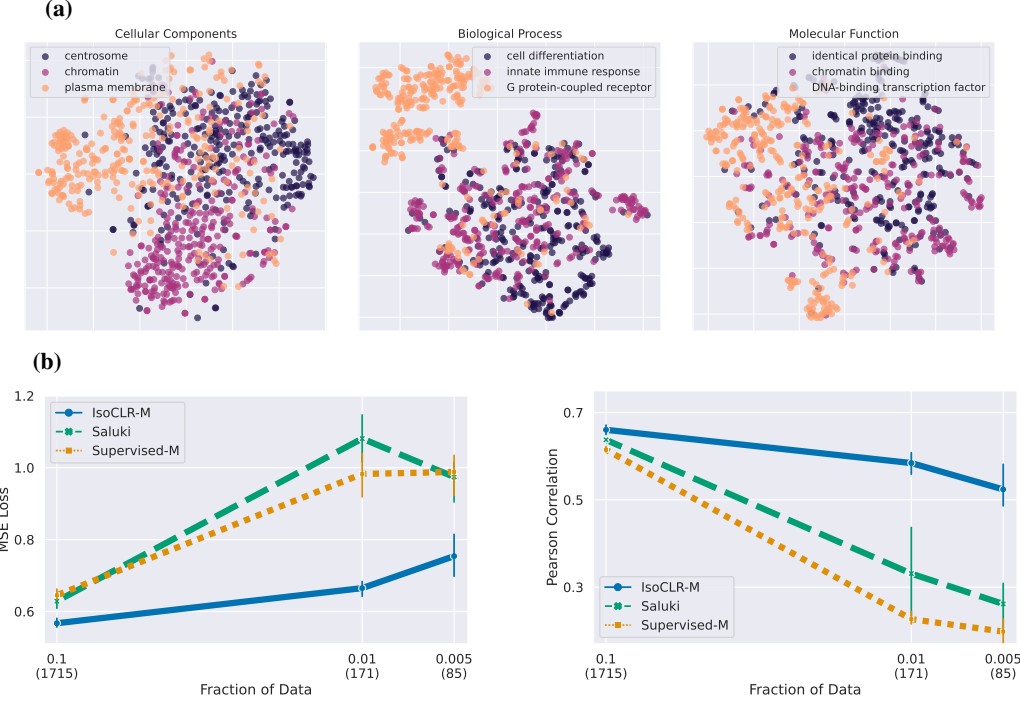

Figure 3: **(a)** Visualization of the learned latent representations with stochastic neighbor embedding. Each dot is an RNA transcript from a unique gene colored by the correspondingly annotated gene ontology. **(b)** Data sub-sampling analysis demonstrating IsoCLR's strong performance in the low data regime. Confidence intervals were computed using standard deviation over three random seeds.

with matched architecture. We also evaluate its performance against a published method for the RNA half-life prediction, Saluki (Agarwal & Kelley, 2022). Experimental details are described in Appendix A.3. We find that the fully fine-tuned IsoCLR model matches the performance of Saluki on the RNA half-life task (Table 3). Furthermore, IsoCLR outperforms fine-tuned HyenaDNA models on both prediction tasks. Other baseline SSL methods such as DNA-BERT2 and RNA-FM have limited input context windows, and cannot be easily applied to these tasks. We observe that scaling the models results in performance degradation, but note that IsoCLR still significantly outperforms baselines with similar parameter counts. Due to high amounts of noise in mean ribosome load data, other self-supervised methods have difficulty generalizing to the mean ribosome load task.

To simulate downstream tasks for which there is a lack of experimental data, we perform fine-tuning on RNA HL prediction where only a subset of the original training data set is available. We observe that supervised methods are ineffective in this regime, while IsoCLR maintains competitive performance (Figure 3b). At 10% and 1% of data used, we observe significant differences between supervised and self-supervised methods. Evaluating using the Pearson correlation coefficient, the gains are even more stark when using only 0.5% of training data. These results demonstrate that Iso-CLR makes advancements with the eventual goal of few-shot learning on downstream tasks where data is insufficient to train supervised methods.

### 4.3 ABLATIONS: SPLICING AUGMENTATIONS ARE KEY

Finally, we investigate the IsoCLR augmentations that contribute towards good performance. We find that sampling splicing isoforms from the same gene is the primary driver of performance (Table 4). Homology, and masking a small percentage of the input sequence provide small gains in overall performance. Homologous gene mapping can be interpreted as removing wrong negatives from our training set, where duplicated genes will have highly similar sequences. Masking the input space can also be thought of as a regularization method which has recently been shown as vital for learning effective representations with contrastive learning (Ben-Shaul et al., 2023).

| Augmentations | Half-life Human MSE | Half-life Mouse MSE | GO AUC |
|---|---|---|---|
| Splice + Homology + Mask + 6t | 0.57 | 0.62 | 0.85 |
| Splice + Homology + Mask | 0.73 | 0.74 | 0.84 |
| Splice + Homology + 6t | 0.58 | 0.70 | 0.84 |
| Splice + Mask + 6t | 0.58 | 0.65 | 0.85 |
| Mask + 6t | 0.88 | 0.86 | 0.77 |

Table 4: Ablation table demonstrating that splicing is the key contributor. Linear probing results were generated by performing gradient descent on a linear layer.

## 5 DISCUSSION

In this work, we demonstrate that by minimizing the distance between mature RNAs generated through gene duplication and alternative splicing, we are able to generate representations useful for RNA property prediction tasks. The pre-training is especially helpful in low data regimes when there are 200 or fewer data points with labels. These situations arise in molecular biology applications, especially in therapeutic domains where compound manufacturing and experiments can be expensive. We demonstrate that self-supervised pre-training is an approach for addressing data efficiency challenges present in genomics and scaling to additional species can be one strategy to generate additional pre-training data.

Previous self-supervised works for genomic sequence property prediction have focused on reconstruction objectives like masked language modeling or next token prediction (Ji et al., 2021; Dalla-Torre et al., 2023). As previously discussed, most genomic positions under little to no negative selection, and are not very informative. Thus, predicting the corresponding tokens introduces little new information to the model. In this work, we instead choose to utilize a stronger inductive bias, minimizing the distance between functionally similar sequences. By relying on a more structured objective, we are able to outperform models that are multiple orders of magnitude larger.

An important question to assess is why we expect that minimizing distances between RNA isoforms would be helpful at all for seemingly unrelated phenotypes like RNA half-life prediction. One hypothesis is that alternative splicing and gene duplication preserve essential RNA regions. Through the contrastive pre-training procedure, we identify these essential regions. By utilizing decoupled contrastive learning, diverse sequences are pushed apart, thus uniformly distributing over the latent space (Yeh et al., 2021). Through encoding these invariances, we find that IsoCLR is able to learn more complicated RNA properties such as cellular component localization and RNA half-life.

A possible limitation of our approach is that by minimizing the representational distance between related sequences, we remove important signals for predicting certain properties. Are there property prediction tasks for which our inductive bias is actually detrimental compared to a randomly initialized model? For RNA half-life, Spies et al. (2013) demonstrated that in more than 85% of genes, isoform choice has no statistically discernible effect. There are other processes for which it is widely considered that alternative splicing is important, such as the development of neurological tissues. [2]

## 6 CONCLUSIONS

In this work, we propose a novel, self-supervised contrastive objective for learning mature RNA isoform representations. We show that this approach is an effective strategy to address two major challenges for cellular property prediction: data efficiency, and model generalizability. We demonstrate that IsoCLR representations are effective in the low data setting, paving the path to true few-shot learning for RNA property prediction. Finally, fine-tuning IsoCLR matches the performance of supervised models beating out other self-supervised methods.

---

[2]There are only 2000 differentially spliced exons across neurological cell lines relative to the 800,000 exons in the human transcriptome (Frankish et al., 2021; Irimia et al., 2014)

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

## A   APPENDIX

### A.1   BIOLOGY PRIMER

We utilize two molecular processes for constructing positive pairs between sequences: splicing and gene duplication. RNA splicing, a process for assembling mature RNA from precursor RNA (pre-RNA). At a high level, splicing involves the removal of non-coding regions, called introns, from the precursor RNA molecule and then joining together the remaining coding regions, called exons, to create the final RNA molecule (Baralle & Giudice, 2017). Importantly, alternative splicing is a prevalent phenomenon in which different combinations of exons can be joined together, leading to the production of multiple RNA isoforms from a single gene. This process greatly increases the diversity of proteins that can be generated from a limited number of genes. The second process is gene duplication and speciation events which generate homologous genes. At a high level, homologous genes are those found in different organisms that have descended from a common ancestral gene. These genes are related by virtue of their shared ancestry and often retain similar functions, structures, or sequences.

### A.2   LINEAR PROBE EXPERIMENTAL DETAILS

In this section, we describe the experimental procedure to evaluate linear probing results.

Given the RNA HL and MRL datasets, we performed a 70-15-15 data split. The data sequences are then embedded by the various self-supervised learning (SSL) models and then aggregated to form a vector which is inputted into a linear regression model.

For IsoCLR, we simply take the mean of the embeddings. For HyenaDNA, we take the mean and max of the embeddings, as well as the last hidden state in the output sequence. Other SSL methods could not handle the length of the input sequences. Thus, when input sequences exceeded the allowable context window, each sequence was chunked to the maximum length allowed by a model, and the embedded chunks were averaged to obtain the embedding for the whole sequence.

After obtaining embedding vectors, we used the scikit-learn implementation of linear regression to perform the linear probes of the embeddings for the downstream tasks. Using the validation split, we also investigated regularizing the regressor (e.g. ridge regression), but found it did not improve performance.

### A.3   FINE-TUNING EXPERIMENTAL DETAILS

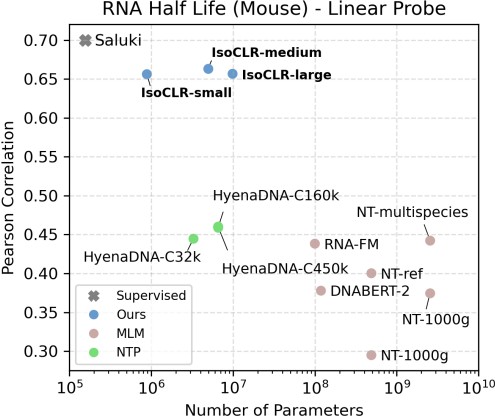

Figure 4: Comparing linear probing performance for self-supervised methods on RNA half-life mouse data.

We fine-tune IsoCLR by first initializing most of the model with weights from pre-training, the penultimate two layers with random initialization, and the final layer with zero init. We don't apply

| Linear Probe GO performance | Cross Entropy Loss | ROC AUC |
|---|---|---|
| IsoCLR Medium (5m) | **0.263** ± 0.01 | 0.84 ± 0.01 |
| DNABert (110m) | 0.316 ± 0.01 | 0.77 ± 0.01 |
| DNABert2 (117m) | 0.315 ± 0.01 | 0.75 ± 0.01 |
| Random Init Linear probe | 0.82 ± 0.18 | 0.55 ± 0.01 |

Table 5: Performance of IsoCLR representation on a 10 class multi-label linear probing task. The classes are gene ontology categories from the molecular function GO hierarchy.

any weight decay to weights that were initialized from pre-training while the final three layers have an l2 weight decay term of 1e-5. We fine-tune on downstream tasks using the Adam optimizer with a learning rate of 0.01. We apply exponential learning rate decay with a factor of 0.95. The models are trained with a single Nvidia T4 GPU in a mixed precision setting.

HyenaDNA models initialized with fine-tuning head. We used the suggested fine-tuning hyperparameters. This includes using the AdamW optimizer with a learning rate of 6e-4 and a weight decay term of 0.1. Models were trained on Nvidia T4 GPUs with a batch size of 28 for the HyenaDNA-tiny and a batch size of 8 for HyenaDNA-small. Models were stopped early based on validation MSE using an epoch patience of three. The runs were repeated using different random initializations to generate confidence intervals.

### A.4 ADDITIONAL COMPARISONS FOR ISOCLR REPRESENTATIONS

To test IsoCLR's latent space semantic interpretability, we take a single transcript from every single human gene and annotate it with gene ontology terms from the three main hierarchies: biological processes, cellular components, and molecular function. We subset the gene ontology terms to third from the root to identify a broad and yet high-level number of functions. From the subset, we select the three most common gene ontology terms to use for the visualization. We also compare the representations to two baselines: a stochastic neighborhood embedding with randomly initialized labels from IsoCLR representation, and a supervised model with a matched architecture trained to predict RNA half-life (van der Maaten & Hinton, 2008) 6. We observe that the supervised embedding also produces a distinct cluster for the biological process go hierarchy confirming the unique structure of g protein-coupled receptors. Upon visual inspection, however, the clusters from the supervised model are less separated compared to the IsoCLR representation. For example, we observe a distinct cluster for sequences associated with the centrosome function in the IsoCLR representation, whereas, for the supervised model, the samples are interspersed throughout the representation 6.

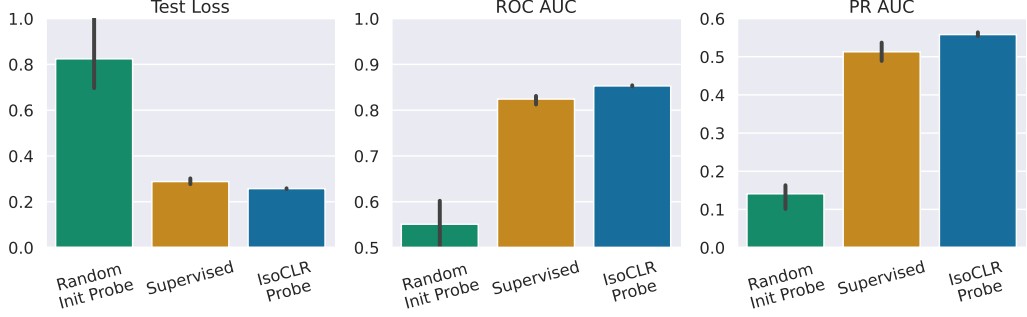

Figure 5: Gene ontology multi-label classification comparison with the supervised model. This is a ten-class multi-label classification task for the molecular function gene ontology category.

To quantitatively validate IsoCLR latent space with gene ontology, we perform linear probing over a 10 class multi-label classification task. Each class corresponds to a gene ontology term, and the samples are RNA sequences with corresponding GO labels. We find that performing linear probing on IsoCLR embeddings exceeds performance of supervised models trained with full fine-tuning (Figure 5, Table 5).

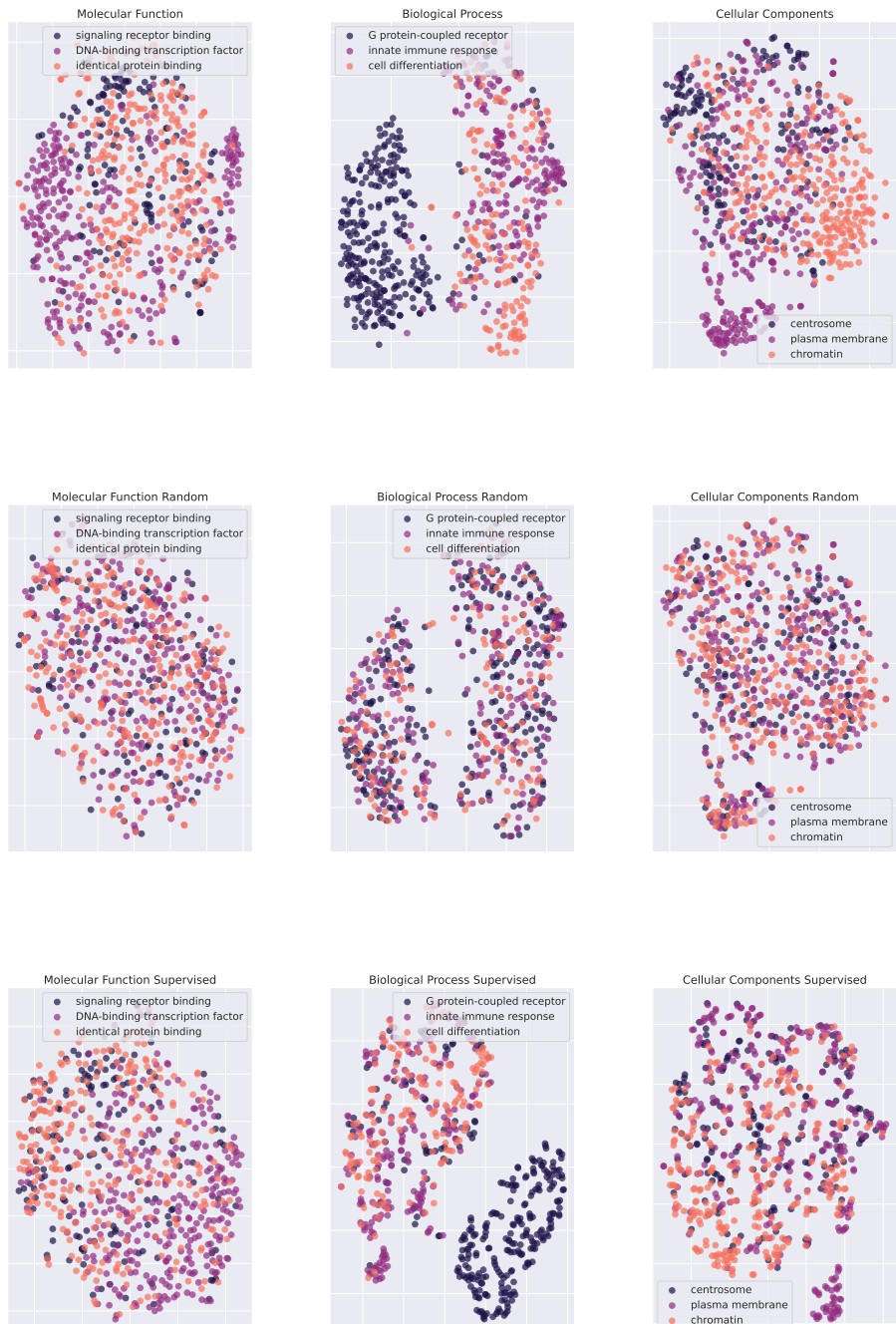

Figure 6: Comparison of embeddings generated through various methods. The first row is generated with the IsoCLR embedding. The second row is generated using random labels from IsoCLR embeddings. The third row is generated using a supervised model trained on RNA half-life.

### A.5 UTILIZED COMPUTE

We train the contrastive model on 4 a40 GPUs for 1000 epochs and a batch size of 1024. This amounts to 40 hours of computing time, however, this can vary depending on the machine due to extra startup time from pre-emption. RNA half-life training takes 2 hours and was done on t4v2 GPUs. We experimented with different hyperparameter choices for contrastive training during which we would train the model on one GPU with a smaller batch size. It is difficult to exactly estimate the number of runs performed but it is on the order of 100 which amounts to around 400 hours of compute.

### A.6 BROADER IMPACT

As with most technologies this work can be used for different purposes. We hope IsoCLR can be used to further improve cellular property prediction and inspire others to consider self-supervised learning methodologies in the genomics domain. A potentially dangerous application of IsoCLR is training the model to learn a semantically meaningful space of viral RNAs. RNA viruses can potentially be a biological weapon that could be misused. Similarly, an improved understanding of RNA viruses can help virologists manufacture effective vaccines and predict evolutionary trends to combat deadly diseases like the flu.

