# OpenReview forum: "Splicing Up Your Predictions with RNA Contrastive Learning"
_ICLR.cc/2024/Conference — Submitted to ICLR 2024_

### Official Review · Reviewer_ZWCi · 2023-11-01

**Soundness:** 3 good
**Presentation:** 3 good
**Contribution:** 2 fair
**Rating:** 5
**Confidence:** 4

**Summary:**

This paper proposes a contrastive learning approach to learn RNA representations that can be employed in downstream tasks. Based on functional similarity between a) alternatively spliced RNA and b) homologous gene, the authors build positive sample pairs through these two similarities. A contrastive training objective adapted from SimCLR is then used to train an RNA encoder, along with projection modules. Experiments are conducted on 3 downstream tasks: 1) RNA half-life and 2) Mean ribosomal load and Gene ontology prediction by training a layer on top of frozen representation. The authors showed that their approach outperform other pretrained represrentations in these tasks and is especially effective in the low data regime.

**Strengths:**

+Strong empirical performance of approach versus baselines
+The paper is easy to read and follow
+Potential application of approache in biomedicine

**Weaknesses:**

The key novelty of the paper seems to hinge on using functional similarity between a) alternatively spliced RNA and b) homologous genes to create positive sample pairs for contrastive learning. There is little innovation in the machine learning aspect (i.e. novel algorithm or significant change from SimCLR), which might make this more suitable for a biomedicine-focused audience rather than the general machine learning community.

**Questions:**

What is the key difference between this approach and existing contrastive objective such as SimCLR apart from exploiting the functional similarity between a) alternatively spliced RNA and b) homologous genes?

==Post-Rebuttal==
I appreciate the authors' response and decided to keep my score.

---

> ### Author Response · Authors · 2023-11-22
>
> We thank you for commending the clarity of our paper and relative improvement over baseline methods. We have summarized our thoughts about novelty in the top-level comment and highlighted the information weighting of samples proportional to their informativeness in the absence of a view-generating function. We respectfully disagree that this is not of interest to the ICLR community given the growing interest in ML applications for biological sciences but appreciate the suggestion and can aim to describe our approach in more general terms that can be relatable to researchers in related domains.

---

### Official Review · Reviewer_qo8f · 2023-11-01

**Soundness:** 2 fair
**Presentation:** 3 good
**Contribution:** 2 fair
**Rating:** 5
**Confidence:** 4

**Summary:**

This paper introduces a contrastive learning-based method for an RNA pre-trained model. It proposes to utilize functional similarities between sequences generated through alternative splicing and gene duplication as positive samples for contrastive learning and train models on it. It performs linear probing on 3 datasets to show that the proposed method performs well against baselines.

**Strengths:**

- The idea of utilizing functional similarities between sequences generated through alternative splicing and gene duplication as positive samples for contrastive learning is sound.
- The paper is well written and easy to follow.

**Weaknesses:**

- The empirical analysis is not convincing.
    - The number of datasets is limited. Only three datasets are used, among which the performance on Gene ontology is only partially reported in the Appendix.
    - What's the point of learning good RNA representations if all your downstream tasks can be solved with standard fine-tuning? In NLP, better sentence embedding can directly be used to amplify retrieval-based QA systems or retrieval-based text generation. But in your application, I do not see the better embeddings helping your downstream applications.
    - If I understand correctly, the proposed method works comparably with `Saluki` in full fine-tuning while performing worse in linear probing than it. It also has more parameters than it. What is the benefit of your model over `Saluki`?
    - How do you use the pre-trained DNA models to solve the RNA prediction tasks? Do you replace all the `U` with `T` and feed them to the DNA pre-trained models? If this is the case, the comparison between the contrastive training target and MLM/NTP is unfair since the models are trained on different corpus.
    - If I understand correctly, the proposed model and `Saluki` utilize more information than the sequence itself. If this is true, then the comparison with the baselines is unfair.
    - Why do you present the results on linear probing with models like RNA-FM, NT, and DNA-BERT2 while skipping them in the model fine-tuning? If they can generate embeddings for linear probing on the datasets, you should be able to fine-tune them on the tasks, too. The models, except for NT-2.5b, are not too large to be fine-tuned on consumer GPUs.

**Questions:**

Please see the Weaknesses section for my questions. Thanks.

---

> ### Author Response · Authors · 2023-11-22
>
> Dear reviewer, thank you for engaging in a thorough evaluation of our paper.
>
> 1. > The number of datasets is limited
>
> We aim to improve on this in a further iteration, however, we believe the substantial improvement observed in linear probing results over previous methods has merit.
>
> 2. > What's the point of learning good RNA representations if all your downstream tasks can be solved with standard fine-tuning?
>
> First, we’d like to point the reviewer to Figure 3b in which we demonstrate IsoCLR’s significant improvements in low data availability prediction setting over supervised methods. In biological sciences lower throughput experiments are more common compared to other domains such as language and vision underscoring the utility of this result.
>
> In addition, we demonstrate IsoCLR’s improvement over supervised models with matched architectures in Table 3. These results demonstrate the general utility of our methods highlighting the potential for improvement across a diversity of architectures.
>
> 3. > What is the benefit of your model over Saluki?
>
> Our linear probing results in Table 2 / Figure 1 compare IsoCLR’s linear probing results to full model fine-tuning. For linear probing we do not have to train the full model, optimizing only the last layer to achieve 85% of the Saluki full fine-tuning performance.
>
> This is interpreted as a successful finding in transfer-learning as the original SimCLR paper also does not outperform the fully supervised ResNet in linear probing regime (they match the performance of the 4x smaller model when restricting training to 90 epochs vs 1000 epochs https://arxiv.org/pdf/2002.05709.pdf figure 7).
>
> In addition, we’d like to point out that we perform significantly better than the existing self-supervised methods.
>
> 4. > Pre-trained DNA models … are trained on different corpus
>
> Indeed, for DNA pre-trained models we replace U->T to create a shared vocabulary between RNA and DNA models. While that in itself does not alter the training distribution (since no information is lost), the DNA models are trained on a superset of our corpus. The distribution shift occurs due to a lack of introns sequences separating exonic regions. We can further clarify this practice in a future revision, underlying the need for RNA-specific models.
>
> We do compare against RNA-FM and find that we outperform the existing RNA self-supervised method.
>
> This observation indicating the difficulty of utilizing DNA models for RNA property prediction tasks underlines the importance of training an RNA-specific self-supervised model.
>
> 5. > The proposed model and Saluki utilize more information than the sequence itself
>
> Most of the methods we compare against are designed for DNA tasks and so do not possess this RNA-relevant information. We did consider this and wanted to understand whether our performance stems from additional information or the pre-training procedure so we conducted an ablation study. We confirm that our smallest model outperforms all the other self-supervised learning methods on both RNA half-life tasks and Gene ontology and can be found under Splice + Homology + Mask row in Table 4. We outperform the other models on the mean ribosome load task as well which we can add to the appendix.
>
> 6. > Why do you present the results on linear probing .. while skipping them in the model fine-tuning
>
> The existing models have a truncated context window relative to the length of RNA sequences. Most methods have 512-1024 context window length whereas to capture 95% of RNA sequences our context length is 12,000 nucleotides in length. During the evaluation of linear probing, we perform a concatenation of average pooling and max pooling to address this issue. It is unclear how to perform this in full-model fine-tuning as the existing context window is between 5-10% of the overall sequence length. We did not perform this experiment since we expected poor results given our margin of performance relative to HyenaDNA. We’ve noted the challenges associated with short context length models in section 4.2 but we will aim to further clarify this in text.
>
> Thank you for engaging with us in the review process, and we would appreciate knowing if our responses clarified your questions.

---

> > ### Comment · Reviewer_qo8f · 2023-11-22
> > **Thanks for your response**
> >
> > Thanks for your response. It solves some of my concerns so I raise my score from 3 to 5. Though this manuscript is not ready for publish for now IMO, I think it’s on a decent direction and will be an impactful work after more interactions.

---

### Official Review · Reviewer_rpSM · 2023-11-01

**Soundness:** 2 fair
**Presentation:** 2 fair
**Contribution:** 2 fair
**Rating:** 3
**Confidence:** 4

**Summary:**

The paper proposes IsoCLR to learn RNA representations by contrastive learning. Splicing as the augmentation is identified as the key for the success.

**Strengths:**

- The paper is easy to follow.
- The method is simple, yet effective in the studied tasks.

**Weaknesses:**

- The work has limited novelty and contributions. It simply applies the contrastive objective to the RNA sequence learning, with the major contribution as identifying an effective augmentation method.
- The work narrows down to learn representations of RNA sequences that mainly can be used for property prediction, making the work less interest and has less impact.
- For example, RNA-FM (which was compared to isoCLR in experiments) demonstrate its effectiveness in structural-related prediction (secondary structure predictions, RNA contact predictions, 3D distances, etc.), which is a more crucial aspect in the related field. The current submission is mostly focused on relatively easier tasks, which is not sufficient to verify the effectiveness of learned representations.

**Questions:**

N/A

---

> ### Author Response · Authors · 2023-11-22
>
> Thank you for a detail-oriented review. Given the overall reviewer sentiment, we have opted not to run experiments for RNA structure prediction for this submission. We agree that including structural prediction tasks will make our evaluations more comprehensive.

---

### Author Response · Authors · 2023-11-22
**Top level comment**

Dear reviewers,

Thank you for taking the time to engage with our work. We believe that identifying contrastive learning views in a new domain is non-trivial and technically interesting. In addition, our setting is distinct from traditional contrastive works since the underlying view-generating function is unknown. Instead, we observe a variable number of views per sample and we introduce a sample informativeness weighting proportional to the number of views observed. This modification to the traditional loss can be of use in other domains with no view generation function available. The weighting function is described by

\begin{equation}
	w_i = log(t_{i} + c)  \frac{T}{\Sigma^N_{k=1} log(t_{k} + c)},
\end{equation}

Where \(t\) is the number of transcripts per gene set, \(T\) corresponds to the total count of transcripts in the dataset, and \(c\) is a constant.

In addition, we highlight important evidence that existing unsupervised ML approaches for genomics are lacking. Although we believe this is an important work especially as the intersection of genomics and ML grows larger, we respect the reviewers' guidance about the suitability of our work for ICLR.

As reviewers have pointed out there isn’t a substantial amount of work applying machine learning to RNA (and genomics broadly), we see this work as an evaluation of successful methods from other domains. In particular, we provide biological intuition and experimental validation for why masked language modeling and next token prediction approaches have been less impressive in genomics domains compared to NLP and suggest an alternative approach for representation learning in genomics.

---

### Meta-Review · Area_Chair_kM8J · 2023-12-06

**Metareview:**

The reviewers were generally unanimous that there might just not be enough methodological novelty beyond the application of relatively standard ideas in contrastive learning (with some small twists) to a new domain. While the authors make the point that applying these ideas in a new domain is non-trivial, machine learning research venues typically expect at least *some* level of methodological innovation. While the bar is certainly lower for applications papers in interesting domains, there wasn't substantial agreement that this bar was met here.

**Justification For Why Not Higher Score:**

The reviewers were generally in agreement on the methodological novelty component piece. I would recommend that the authors spend more time in the paper emphasizing the differences between existing approaches to contrastive learning.

**Justification For Why Not Lower Score:**

N/A

---

### Decision · Program_Chairs · 2024-01-16

Reject